# Influenza A and D Viruses in Non-Human Mammalian Hosts in Africa: A Systematic Review and Meta-Analysis

**DOI:** 10.3390/v13122411

**Published:** 2021-12-02

**Authors:** Annie Kalonda, Marvin Phonera, Ngonda Saasa, Masahiro Kajihara, Catherine G. Sutcliffe, Hirofumi Sawa, Ayato Takada, Edgar Simulundu

**Affiliations:** 1Department of Biomedical Sciences, School of Health Sciences, University of Zambia, Lusaka 10101, Zambia; 2Department of Disease Control, School of Veterinary Medicine, University of Zambia, Lusaka 10101, Zambia; marvinphonera@gmail.com (M.P.); nsaasa@gmail.com (N.S.); h-sawa@czc.hokudai.ac.jp (H.S.); atakada@czc.hokudai.ac.jp (A.T.); 3Africa Centre of Excellence for Infectious Diseases of Humans and Animals, School of Veterinary Medicine, University of Zambia, Lusaka 10101, Zambia; 4Machinga Agricultural Development Division, Ministry of Agriculture, Irrigation and Water Development, Liwonde 303110, Malawi; 5Division of Global Epidemiology, International Institute for Zoonosis Control, Hokkaido University, N 20 W10, Kita-ku, Sapporo 001-0020, Japan; kajihara@czc.hokudai.ac.jp; 6Department of Epidemiology, Johns Hopkins University Bloomberg School of Public Health, Baltimore, MD 21205, USA; csutcli1@jhu.edu; 7Division of Molecular Pathobiology, International Institute for Zoonosis Control, Hokkaido University, N 20 W10, Kita-ku, Sapporo 001-0020, Japan; 8International Collaboration Unit, International Institute for Zoonosis Control, Hokkaido University, N 20 W10, Kita-ku, Sapporo 001-0020, Japan; 9Global Virus Network, 725 W Lombard Street, Baltimore, MD 21201, USA; 10Macha Research Trust, Choma 20100, Zambia

**Keywords:** animal influenza, influenza A virus, influenza D virus, Africa, prevalence, seroprevalence

## Abstract

We conducted a systematic review and meta-analysis to investigate the prevalence and current knowledge of influenza A virus (IAV) and influenza D virus (IDV) in non-human mammalian hosts in Africa. PubMed, Google Scholar, Wiley Online Library and World Organisation for Animal Health (OIE-WAHIS) were searched for studies on IAV and IDV from 2000 to 2020. Pooled prevalence and seroprevalences were estimated using the quality effects meta-analysis model. The estimated pooled prevalence and seroprevalence of IAV in pigs in Africa was 1.6% (95% CI: 0–5%) and 14.9% (95% CI: 5–28%), respectively. The seroprevalence of IDV was 87.2% (95% CI: 24–100%) in camels, 9.3% (95% CI: 0–24%) in cattle, 2.2% (95% CI: 0–4%) in small ruminants and 0.0% (95% CI: 0–2%) in pigs. In pigs, H1N1 and H1N1pdm09 IAVs were commonly detected. Notably, the highly pathogenic H5N1 virus was also detected in pigs. Other subtypes detected serologically and/or virologically included H3N8 and H7N7 in equids, H1N1, and H3N8 and H5N1 in dogs and cats. Furthermore, various wildlife animals were exposed to different IAV subtypes. For prudent mitigation of influenza epizootics and possible human infections, influenza surveillance efforts in Africa should not neglect non-human mammalian hosts. The impact of IAV and IDV in non-human mammalian hosts in Africa deserves further investigation.

## 1. Introduction

Influenza viruses (IVs) are enveloped, single-stranded RNA viruses with segmented genomes containing 7–8 gene segments. They belong to the family *Orthomyxoviridae* and consist of four genera: *Alphainfluenzavirus* (Species: *Influenza A virus* (IAV)), *Betainfluenzavirus* (Species: *Influenza B virus* (IBV)), *Gammainfluenzavirus* (Species: *Influenza C virus* (ICV)) and *Deltainfluenzavirus* (Species: *Influenza D virus* (IDV)) that are classified according to antigenic variations of their nucleoprotein (NP) and matrix 1 (M1) proteins [1,2,3,4]. The four influenza virus genera differ in host range and pathogenicity and are likely to have diverged evolutionarily at least several thousand years ago [5]. Among these genera, IAVs are the most virulent and are known to cause severe disease. Further, only IAVs pose a significant risk of zoonotic transmission, host switching, and the generation of pandemic IAVs [5].

Wild waterfowl among the orders Anseriformes and Charadriiformes are considered to be the natural reservoirs for IAVs [6]. IAVs are classified into subtypes based on their antigenic and genetic diversity of two surface glycoproteins, haemagglutinin (HA) and neuraminidase (NA). To date 16 HA (H1–H16) and 9 NA (N1–N9) subtypes of IAVs have been detected and circulate in wild waterfowl and poultry [7]. In addition, IAV-like viruses, H17N10 and H18N11, were recently detected in bats from Guatemala and Peru, respectively [8,9]. Some IAV subtypes have crossed species barriers, establishing stable lineages in a wide variety of animals [10,11], for example H1N1 and H3N2 subtypes in humans [12,13], H1N1, H1N2, and H3N2 subtypes in swine [14,15] and H3N8 and H7N7 subtypes in horses [16,17].

Interspecies transmission of IAVs is common among different animal species via direct or indirect contact which may result in the introduction of viruses that are new to the recipient species and which have the potential to cause substantial outbreaks [6]. Whereas most of these interspecies transmission events may not result in onward transmission and establishment in the new host, sustained influenza outbreaks have been reported in poultry and several mammalian species [18]. Of the mammalian hosts, only a limited number are currently recognised as sustaining IAV transmission, and it is not clear what distinguishes these species from those for which influenza has not been reported [18]. However, for IAVs to become established and achieve efficient viral replication in other hosts, they must overcome a variety of species barriers [19]. Such barriers include host innate immune responses, several intracellular factors and recognition of different sialic acid (SA) receptors, α-2,3 and α-2,6 expressed on host cell surfaces of avian and human respiratory epithelia, respectively [20]. The well-known mammalian hosts for which IAVs have established themselves include humans, pigs, horses, seals, mink and dogs. Dogs emerged as important IAV hosts in the 2000s when the H3N8 equine influenza virus (EIV) and the avian virus-like H3N2 strain introduced from horses and birds, respectively, were detected in the United States of America (USA) and Asia [21,22]. Both of these canine influenza viruses have continuously circulated in the dog population since their emergence, increasing opportunities for human exposure to these zoonotic viruses [18].

Apart from reports of IAVs in domestic animals, IAVs of various subtypes have been documented in wild animals though these reports are mainly limited to captive animals. Examples of these introductions include H5N1 IAV infections in leopards and tigers in Thailand [23,24], and the H1N1 virus that caused the 2009 pandemic (H1N1pdm09) in cheetahs in California USA [25] and wild boars in Japan [26]. Infections in wild animals are usually thought of as being opportunistic as they usually arise through the consumption of raw meat containing the virus especially for carnivores, hence limited or no animal to animal transmission occurs. However, a study by Thanawongnuwech et al. [27] in tigers points to a probable horizontal transmission of IAVs in these animals. Although herbivores might be exempt from diet-driven pathogen transmission, sharing common feeding grounds and water sources with the reservoir host could also lead to potential transmission [28].

While IAVs cause mild to severe disease in various animal species, IDV has been associated with bovine respiratory disease complex which is the most economically significant disease of the beef industry with economic losses being attributed to morbidity, mortality, treatment costs, and reduced carcass value [29,30]. IDV was recently discovered in swine with respiratory disease in the USA in 2011 [31]. Since its discovery, serological evidence of IDV has been reported in healthy and symptomatic cattle populations in multiple geographical regions including the USA [1,31,32,33], Europe [34,35,36,37], Asia [38,39] and Africa [40], suggesting that cattle could be the natural reservoir hosts of this new virus [1]. Further, serological evidence of IDV has been reported in small ruminants in the USA [41]. Despite various studies on the virological and serological evidence of IDV, its zoonotic potential and pathogenicity in other hosts including humans remain obscure.

Despite the increasing knowledge of the dynamics of IVs in different avian and non-human mammalian species around the globe, current data in Africa are limited, characterised by patchy surveillance studies, thereby limiting a more comprehensive understanding of the prevalence and circulation of these viruses in non-human mammalian species. Furthermore, the potential public health risk of these viruses arising from the close relationship between non-human mammalian species (pigs, dogs, cats and horses among others) and their owners underscores the need to study their prevalence and circulation in these animals. Therefore, we conducted a systematic review and meta-analysis to investigate the prevalence and current knowledge of IAV and IDV in non-human mammalian hosts in Africa.

## 2. Materials and Methods

### 2.1. Literature Search Strategy

The systematic review and meta-analysis were conducted according to the Preferred Reporting Items for Systematic Reviews and Meta-Analyses (PRISMA) guidelines (Appendix A) [42], Figure 1. Three databases, PubMed, Google Scholar and Wiley Online Library were searched using terms related to IAV and IDV in non-human mammalian hosts (Appendix A). In addition, the World Organisation for Animal Health–World Animal Health Information System (OIE–WAHIS) platform was also used as a data source as it provides direct up-to-date information on animal health situations worldwide. All references located in the searches were imported into Endnote Version 8, a web-based reference manager, and a database for all relevant articles was generated.

### 2.2. Study Selection

All studies identified in the search were assessed, and duplicates removed and checked for eligibility. The studies were initially selected based on the relevance of their titles and abstracts regarding the prevalence and circulation of IAV and IDV in non-human mammalian hosts in Africa. Thereafter, full texts of the remaining articles were screened and those that did not meet the inclusion criteria were excluded.

### 2.3. Inclusion and Exclusion Criteria

The review included all study types on IAV and IDV in non-human mammalian species with the exception of (i) experimental studies, (ii) studies on the development of new diagnostic methods and (iii) vaccine development. Only studies written in English and published between 2000 and 2020 were included in this review and meta-analysis.

Studies excluded from the review included those not published in Africa, those published before 2000 or after 2020, editorials, conference proceedings, review articles, animal experiments, theoretical models, and studies in human and avian species. Studies were further excluded if the diagnostic test was not indicated, had overlapping data with another included study and were excluded from the meta-analysis if the sample size was less than five.

### 2.4. Data Extraction

We extracted study information regarding the author’s name, title and year of publication. Additional information extracted included country/region, study type, animal species, sample type, diagnostic method, sample size, number of positive samples, IAV subtype, strain, vaccination status (important for swine and equids), and premises (indicating where the sample was collected such as farm and slaughterhouse, etc.).

### 2.5. Assessment of Quality and Risk of Bias

We assessed the quality and risk of bias of included studies using a quality assessment checklist (Appendix A) [43,44]. The checklist included ten questions that had a ‘yes’ or ‘no’ answer. A point was scored if the response was ‘yes’ and zero for ‘no’. Overall study quality was categorised as ‘high’ (scores ≥ 8 points), ‘moderate’ (scores 5 to 7 points) or ‘low’ (scores < 5 points). Funnel and doi plots were used to assess publication bias using the LFK index. Based on the LFK index no asymmetry was defined as LFK index values within ±1, minor asymmetry as values exceeding ±1 but within ±2, while major asymmetry as values exceeding ± 2 [45].

### 2.6. Statistical Analysis

Descriptive statistics were used to describe the overall search results, characteristics of included studies and distribution of IAVs and IDV in non-human mammalian hosts using MS Office Excel^®^ 2016. For the meta-analysis, MetaXL version 3.1 (https://epigear.com accessed on 26 July 2021) a tool for meta-analysis in Microsoft Excel was used to pool prevalence and seroprevalences from each study [46]. Seroprevalence was defined as the presence of antibodies against IVs by any serological test while prevalence was defined as the isolation or detection of IVs by culture or reverse-transcriptase polymerase chain reaction. The quality effects model was used to calculate the pooled prevalences and their 95% confidence intervals (CI). The I^2^ was used to assess study heterogeneity and I^2^ values of 25%, 50% and 75% were considered as having a low, moderate and high degree of heterogeneity, respectively [47]. We divided studies into subgroups based on the geographical regions (African Islands, Central, East, Southern and West Africa) and animal hosts to investigate the potential sources of heterogeneity.

## 3. Results

### 3.1. Search Results, Study Selection and Characteristics of the Included Studies

A total of 8785 articles and records were identified, of which 49 (45 articles and four records from OIE-WAHIS) were included in this study (Figure 1 and Appendix A). Of the 49 articles/records included in the systematic review, 29 were included in the meta-analysis. Further, some articles reported datasets from more than one country in Africa. Overall the included articles/records reported data from 19 African countries as shown in Figure 2. The highest number of articles was for studies conducted in West Africa (24/49; 49.0%) with the majority of articles being from Nigeria while the least of articles were for studies conducted in Southern Africa and the African Islands which recorded one study each (1/49; 2.0%). Furthermore, the majority of the articles/records were published between 2011–2020 (40) while only nine articles/records were published between 2000–2010 (Figure 3).

According to the included studies, more studies/records (28 (45.2%)) were conducted in pigs than in any other animal species included in this study (Appendix A). Further, most studies collected serum or both serum and nasal swabs (18 (36.7%)) and used multiple methods (serological or virological methods) (22 (44.9%)) for the identification of IVs (Appendix A). Only four (8.2%) articles that reported on IDVs in Africa were included in this study. Furthermore, most studies did not report whether the animals had an influenza-like illness (ILI), or on their vaccination status (Appendix A).

### 3.2. Assessment of Quality and Risk of Publication Bias of Selected Studies

According to our quality assessment criteria, of the 29 publications included in the meta-analysis, five publications were of high quality, 13 were of moderate quality and 11 were of low quality. Publication bias in studies was measured and detected using the funnel and doi plots. Overall, the funnel and doi plots (Appendix A) showed minor and major asymmetry with the LFK index of 3.52 and 1.48 for prevalence and seroprevalence of IAV in pigs, respectively, and 1.10 for seroprevalence of IDV in non-human mammalian species, demonstrating a potential risk of publication bias among the selected papers.

### 3.3. Distribution of Influenza A and D Viruses and Detection of Antibodies in Non-Human Mammalian Hosts in Africa

The distribution of IAV and IDV in non-human mammalian hosts is depicted in Table 1 and Table 2. West Africa had the highest number of countries (n = 9) with studies or reports on IAV in non-human mammalian species followed by North Africa (n = 4), East Africa (n = 3), African Island (n = 1), Central Africa (n = 1) and Southern Africa (n = 1). All the regions (African islands, Central, East, North, Southern and West Africa) included in this review reported at least virological or serological evidence of IAV in non-human mammalian species (Table 1).

The majority of the studies included in this review provided virological and/or serological evidence of the circulation of H1N1pdm09 in pigs in Africa (Table 1). The countries reporting H1N1pdm09 in pigs included Cameroon, Egypt, Ghana, Kenya, Nigeria, Togo and the Reunion Island. Apart from H1N1pdm09, classical H1N1, H3N2, H1 and H3 viruses were also reported in pigs in Burkina Faso, Kenya, Egypt, Uganda, Ghana and Nigeria (Table 1). The populations of pigs in various studies included piglets, weaners, growers, finishers, sows and boars. Additionally, pigs were either sampled from farms or slaughterhouses, though some articles did not indicate the sources of the pigs sampled. Further virological and serological evidence of H5N1 highly pathogenic avian influenza viruses (HPAIVs) were reported in Egypt and Nigeria, as well as H5N2 and H9N2 viruses in Egypt (Table 1).

Additionally, virological and/or serological evidence of exposure to IAVs was also reported in cats, dogs, rats, olive baboons (*Papio anubis*), equids, bats, spotted hyena (*Crocuta crocuta*), black rhinos (*Diceros bicornis*), wildebeest (*Connochaetes taurinus*) and caracals (*Caracal caracal*) (Table 1). IAV-specific antibodies for H1 and H1N1 were detected in cats and dogs in Kenya [48], H3N8 and other unidentified subtypes in hunting, pet and village dogs in Nigeria [49,50], while antibodies against H5N1 were detected in cats, dogs and rats in Egypt [51] (Table 1). Further, influenza A viral RNA was detected in dogs in Kenya [48]. Equine influenza virus (EIV) subtype H7N7 (EIV-1) and/or, H3N8 (EIV-2), and their respective antibodies were reported in horses, donkeys and mules in Egypt [52,53]. Moreover, antibodies specific for EIV were detected in donkeys, horses and mules in Morocco [54], Sudan [55], Tunisia [56], Mali [57] and in camels in Kenya [58]. Apart from EIVs, H5N1 HPAIV and antibodies against this virus were detected in donkeys in Egypt [59] (Table 1). Furthermore, specific antibodies against H3, H5, H8, H9 and H12 viruses were also detected in wild mammals such as bats in Ghana [60] and IAV A/bat/Egypt/381OP/2017 was detected from Egyptian fruit bats in Egypt [61]. Moreover, a study conducted in Namibia demonstrated the exposure of various wildlife animals such as lions, black rhinos, spotted hyena, wildebeest, caracal, honey badgers and black-backed jackal to various IAVs [28] (Table 1).

Exposure to IDV was reported in East Africa, North Africa and West Africa. IDV antibodies were detected in cattle from Benin [40], Morocco [90] and Togo [40,91], in dromedary camels from Kenya [40] and Ethiopia [92], and in small ruminants from Ethiopia and Togo (Table 2). Further, the review of the literature suggests that IDV has been circulating in Africa since 2012 as evidenced by the antibodies detected in Morocco [40].

### 3.4. Pooled Prevalence, Seroprevalence and Heterogeneity of IAVs in Pigs in Africa

The estimated pooled prevalence of IAV in pigs in Africa was 1.6% (95% CI: 0–5%), *I*^2^ = 98%, *p* < 0.0001 as shown in Table 3 and Figure 4A. African Islands and North Africa had the highest prevalence of 13.2% (95%: 10–16%) and 10.4% (0–100%), respectively, while the lowest prevalence of 0.3% (95% CI: 0–1%) was observed in East Africa. Furthermore, the pooled prevalence of IAV in pigs varied across studies ranging from 0–63% (Figure 4A).

The estimated pooled seroprevalence of IAV in Africa was 14.9% (95% CI: 5–28%), *I*^2^ = 99%, *p* < 0.001 among pigs (Table 3 and Figure 4B). The highest prevalence was recorded in African Islands with 33.2% (95% CI: 31–36%), followed by Central Africa with 27.8% (95% CI: 19–37%), North Africa with 25.8% (95% CI: 0–100%), West Africa with 14.9% (95% CI: 0–41%) and the least was 12.6% (95% CI: 7–18%) for East Africa (Table 3 and Figure 4B). Further, there was a variation in the pooled seroprevalence of IAV in pigs among individual studies ranging from 0–94% as shown in Figure 4.

### 3.5. Pooled Seroprevalence and Heterogeneity of IDV in Non-Human Mammalian Hosts

Of the 29 studies included in the meta-analysis, only four were on IDV. The overall seroprevalence of IDV in non-human mammalian species was 9.9% (95% CI: 0–28%), *I*^2^ = 99%, *p <* 0.001 as shown in Table 4. The seroprevalence of IDV was highest in camels with 87.2% (95% CI: 24–100%) and lowest in pigs with 0.0% (95% CI: 0–2%) (Table 4 and Figure 5).

## 4. Discussion

The main objective of this systematic review and meta-analysis was to investigate the prevalence and circulation of IAVs and IDVs in non-human mammalian hosts in Africa. This review included all studies found in the searched databases which reported data on prevalence, seroprevalence, virus isolation and genome detection rates of influenza A and D viruses in non-human mammalian hosts in Africa between 2000 and 2020. A total of 8785 articles were retrieved from the databases and other sources of which 169 full-texts were screened and 49 were selected and included in this review.

The results of this review and meta-analysis showed that the majority of studies were conducted in West Africa, predominantly from Nigeria. Additionally, our review demonstrated an increase in the number of studies performed after 2011. This increase in the number of studies could be attributed to the heightened interest in IAV in non-human mammalian species, especially swine, after the 2009 H1N1 pandemic. It is also possible that rigorous sampling and reporting of surveillance activities in non-human mammalian species were absent before the pandemic and more surveillance effort was concentrated on the emergence of H5N1 HPAIV as evidenced by numerous studies conducted in avian species [93]. Furthermore, the discovery of the novel IDV virus in swine in the USA and bat influenza in South America in 2011 could also have contributed to the increased number of studies of IVs in non-human mammalian species after 2011. Moreover, more studies were reported in pigs than any other animal species included in this review and meta-analysis.

The present review showed that the predominant IAVs circulating in pigs in Africa from 2000 to 2020 were H1N1 and H1N1pdm09 followed by H3N2 viruses. RNA and antibodies of the H1N1pdm09 virus were the most frequently detected among studies included in this review, suggesting that reverse zoonosis could be a common occurrence in Africa. The H1 subtypes were detected in five regions of Africa namely African Islands [62], Central Africa [63,64,65], East Africa [48,68,71], North Africa [73] and West Africa [65,78,79,82,84,85,86,87], but not Southern Africa, where no reports of studies in pigs were included in this review. While H3 was detected in Central Africa [63], North Africa [73], and West Africa [65,76,78,84,85], it was not detected in African Island, East and Southern Africa. Our findings are in line with those of studies in China [94] and Korea [95,96] which detected H1 and H3 subtypes as being predominant in pigs. Furthermore, our review is in agreement with the general notion that H1N1, H1N2 and H3N1 IAVs are endemic in pigs throughout the world [14,15]. The findings also revealed the circulation of other non-pig-adapted IAV subtypes in apparently healthy pigs including the H5N1 HPAIV clade 2.3.2.1c reported in Nigeria [82], H5N1 clade 2.2.1.2, H5N1 and H5N2 viruses in Egypt [51,72,73]. The detection of viral RNA in apparently healthy pigs in Nigeria is a public health concern as it shows the silent circulation of a potentially zoonotic HPAIV in a country with a large population of pigs reared under intensive and free-range husbandry systems [82]. The other subtype detected in pigs was the H9N2 low pathogenic avian influenza virus reported in Egypt [73]. Similar observations of H5N1 and H9N2 circulation in pigs have been reported in China [94]. The exposure of pigs to avian influenza viruses (AIVs) has been attributed to the increased occurrence of AIV outbreaks in poultry in the two regions (North and West Africa) as well as pigs feeding on dead poultry carcasses or droppings of wild birds, which typically share their food [59,82]. Moreover, the co-circulation of pig adapted IAVs, non-pig-adapted IAVs and AIVs in pigs in Africa raise concern, as this may result in co-infections and possibly the generation of new reassortant viruses with pandemic potential as pigs are recognized to be “mixing vessel” of pandemic influenza virus strains [82].

The results of the meta-analysis showed an estimated pooled prevalence of 1.6% (95% CI: 0–5%) of IAV in pigs in Africa. This finding is comparable to a study in Cambodia which reported a prevalence of 1.5% of IAV in pigs [97] but lower than the 11.7–15.7% and 19.67% reported in Guatemala [98] and Mexico [99], respectively. Further, the meta-analysis demonstrated an estimated pooled seroprevalence of 14.9% of IAV in pigs in Africa. The findings are relatively similar to other studies in Britain and Wales [100], Cambodia [101], and Malaysia [102], which reported an overall seroprevalence of 12–14.9%. In contrast, higher seroprevalences of 30 to ≥50% in Belgium, Germany, Italy and Spain [103,104], 46.1% in Korea [105], 37.7% in Taiwan [106] and 22.8% in the USA [107] have been reported in pigs. The differences observed in prevalence and seroprevalence of IAV in pigs could be attributed to the region where the studies were conducted, the status of the animals (healthy or diseased), age of the animals, type of sample, sample sizes and diagnostic tests used.

The findings also demonstrated the presence or circulation of EIV in camels in Kenya [58], and horses, donkeys and mules in Egypt [52,53,59], Mali [57], Niger [77], Nigeria [83,88] Senegal [77], Sudan [55] and Tunisia [56]. The present review reported the detection of H3N8 and H7N7 antibodies and viral RNA of EIV in horses, donkeys and mules. These two subtypes of IAV have been associated with influenza virus disease in horses [16,17]. Despite the idea that H7N7 may be extinct, our review reported serological evidence of this subtype in Egypt [52] and Nigeria [54]. Further, the horses, donkeys and mules in these two studies were not vaccinated, indicating natural exposure of these equids to EIVs. Therefore, this finding may suggest the possible silent or undetected circulation of H7N7 EIV in African equids. In addition, H5N1 HPAIV clade 2.2, sub-clade 2.2.1 was detected from donkeys showing influenza-like illness in Egypt [59] suggesting active infection. Exposure of Egyptian horses and donkeys to H5N1 AIV suggests the susceptibility of equids to this virus and raises concern regarding the role of equids in the spread of the H5N1 virus to other animal species [59]. Transboundary movement of donkeys, horses and mules has been implicated in EIV infections in West Africa. It has been suggested that herders often use donkeys to transport goods and once infected these animals can carry pathogens between regions and countries due to porous borders [77].

Serological evidence has shown that dogs could be infected with human influenza viruses, and different subtypes of IVs even coexist in dogs [108,109]. The present review demonstrated serological evidence of H1N1, H3N8 and H5N1 IAV in dogs and cats from Nigeria, Kenya and Egypt [48,49,50,51]. These results suggest that IAV could be circulating in household dogs and cats in Africa. Furthermore, pet dogs and cats share the same environment with backyard poultry and are in close contact with their owners, therefore increasing the opportunities for human exposure to these viruses. Therefore, continued surveillance of IAVs in dogs and cats is cardinal to determine the risk posed by canine-derived IAVs to public health.

This review further demonstrated the exposure of African wildlife to IAVs including lions, black rhino, spotted hyena, wildebeest, caracal, black-backed jackal, olive baboons, rats and bats [28,51,60,61,69]. The detection of IAV antibodies or antigens in wild mammals correlates with a study in Thailand and China that reported the detection of H5N1 HPAIV in leopards and tigers [23,110] though the present review did not determine whether the strains identified serologically represent low- or highly pathogenic IAV strains. The exposure of wild mammals to IAVs could be attributed to the consumption of contaminated meat in carnivores or contaminated water or feeding grounds for herbivores [28]. For example, captive carnivores, including tigers, leopards, dogs, cats, and raccoons, have been observed with influenza symptoms after consumption of contaminated meat [111,112,113].

Bats are reservoir hosts of many zoonotic viruses, such as the severe acute respiratory syndrome (SARS) coronaviruses, Middle East respiratory syndrome coronavirus (MERS), Nipah and Hendra viruses among others, which can cause severe disease and significant mortality in humans [114,115]. In contrast to known bat influenza viruses (H17N10 and H18N11), this review found a report of a novel H9N2-like virus (A/bat/Egypt/381OP/2017) which was detected in oral and faecal swab samples collected from Egyptian fruit bats in a densely populated agricultural area in Egypt [61]. We also found studies reporting serological evidence of IAV subtype H3, H5, H8, H9 and H12 in straw-coloured fruit bats in Ghana [60]. The H9N2-like virus is thought to be transmitted through the faecal-oral route which suggests opportunities for human exposure to this kind of virus through bat faeces and saliva on contaminated fruits [61,116]. The virological and serological detection of IAV in wild mammals highlights the risk that IAVs pose to many mammals, including humans, as their transmission dynamics and host ranges are unclear.

Studies around the globe have reported the circulation of IDV in either healthy or sick cattle, small ruminants and swine from China, France and the USA [30,32,34,117,118]. This review and meta-analysis demonstrated the presence of IDV specific antibodies in cattle from Benin, Morocco and Togo [40,90,91], camels from Ethiopia and Kenya [40,92] and small ruminants from Ethiopia and Togo [40,91,92]. However, no viral RNA of IDV was detected, possibly due to the absence of active infection in the animals during the period of sampling, the limited number of samples collected in each study, and the limited number of studies conducted in Africa. The estimated pooled seroprevalence of IDV varied widely among different host species ranging from 0.0% (95% CI: 1–2%) in pigs to 87.2% in camels (95% CI: 24–100%) with an overall seroprevalence of 10% (0–28%). With cattle being considered to be the reservoir host of IDV, it was intriguing that the highest seroprevalence was observed in dromedary camels. This suggests that these animals could be susceptible to IDV infection and are worthy of monitoring to better understand their role in the epidemiology of IDV. The seroprevalence observed in cattle, small ruminants and pigs in Africa was lower than that reported in the USA, France and Japan [32,33,117,119,120]. While studies in other parts of the world have reported IDV in pigs [36,117], the findings of this review reported a zero seroprevalence rate. This could be attributed to the small sample size of the included study which was the only study investigating IDV in pigs in this review and meta-analysis. This calls for more IDV studies to be conducted in Africa to ascertain the true picture of IDV circulation in pigs. Furthermore, the serological data of IDV in cattle, camels and small ruminants is likely to reflect natural infection as there is no IDV vaccination in place [91].

The potential limitations of this review include language restriction due to papers published only in English, the large heterogeneity and publication bias observed across studies, sub-regions and host species. Studies were conducted in a limited number of African countries, with West Africa being overrepresented. Reasons for this discrepancy are unclear but may reflect limited technical and financial capacity, underreporting, with few articles being published in journals accessible online, and animal influenza not being a research priority for some regions of the continent. Therefore, more studies on IAVs and IDVs in non-human mammalian species need to be conducted in Africa to identify the annual and seasonal patterns in prevalence and seroprevalence as well as to monitor the evolution and circulation of these viruses, thus assisting in preparing for potentially emerging influenza viruses of animal origin in humans.

## 5. Conclusions

This review and meta-analysis found that IAVs and IDVs are currently circulating in non-human mammalian hosts in Africa with an estimated pooled prevalence and seroprevalence of 1.6% and 14.9% in pigs, respectively, while the seroprevalence of IDV was estimated to be 9.9%. Pig and non-pig adapted IAVs are currently circulating in Africa with H1N1 and H1N1pdm09 predominating. Furthermore, virological and/or serological evidence of H3N8 and H7N7 in equids, H1N1, H3N8 and H5N1 in dogs and cats were reported. Therefore, the circulation of these viruses in non-human mammalian hosts underscores the need for continued IAV surveillance in different animal species to evaluate and possibly mitigate potential threats that these viruses may pose to public health, wildlife and the livestock industry. This may help develop new surveillance plans and determine high-risk regions. Further, we recommend more research to be conducted across Africa to ascertain the impact of influenza A and D viruses in non-human mammalian hosts in Africa.

## Figures and Tables

**Figure 1 viruses-13-02411-f001:**
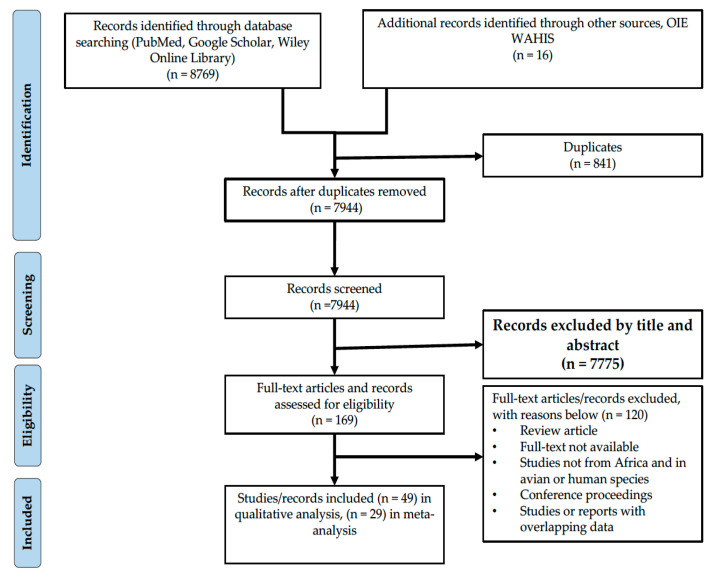
PRISMA flow diagram of the selection process used to determine eligible studies.

**Figure 2 viruses-13-02411-f002:**
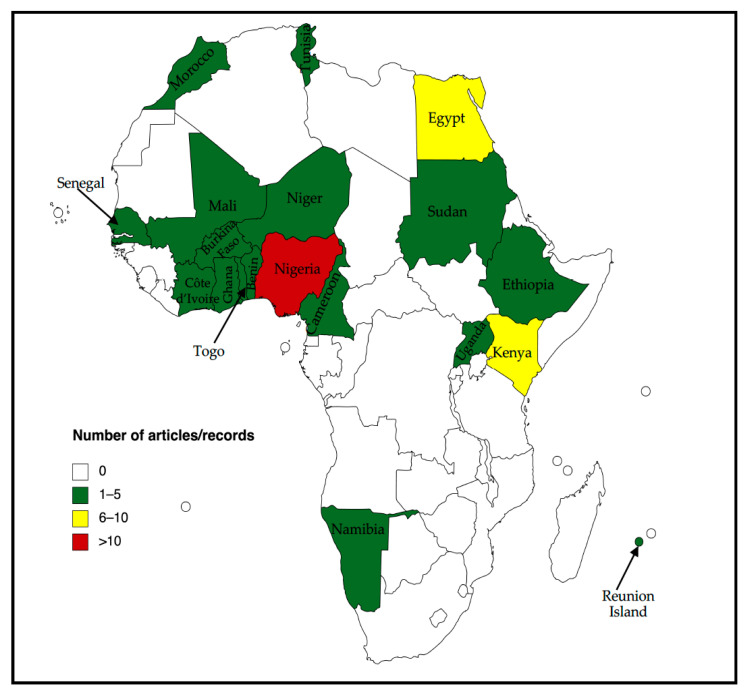
Map of Africa showing the distribution of the number of articles/records (n = 49) included in the review. Some articles reported data from several countries. The map was created online at https://mapchart.net/ (accessed on 30 November 2021).

**Figure 3 viruses-13-02411-f003:**
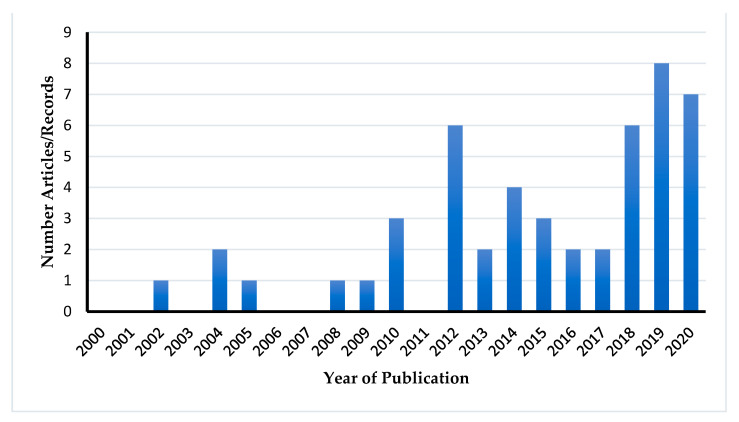
Number of publications included in the present study from 2000 to 2020.

**Figure 4 viruses-13-02411-f004:**
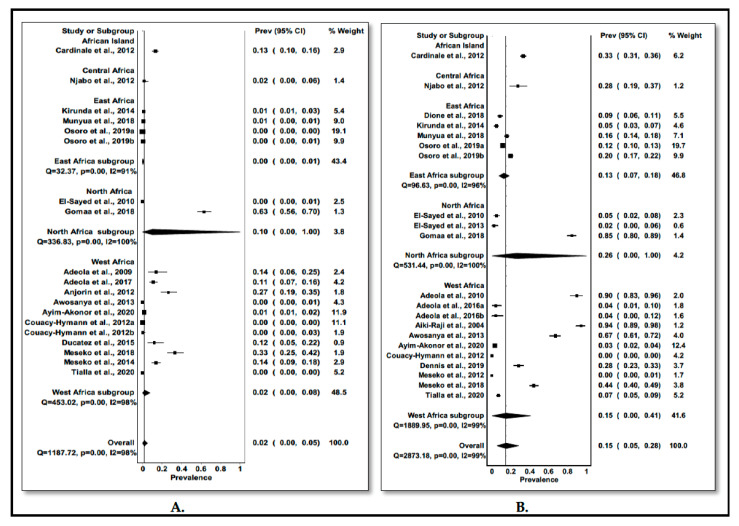
Forest plot of the prevalence and seroprevalence estimates IAV. (**A**). Forest plot of the prevalence estimates of IAV in pigs in Africa by region; (**B**). Forest plot of the seroprevalence estimates of IAV in pigs in Africa by region.

**Figure 5 viruses-13-02411-f005:**
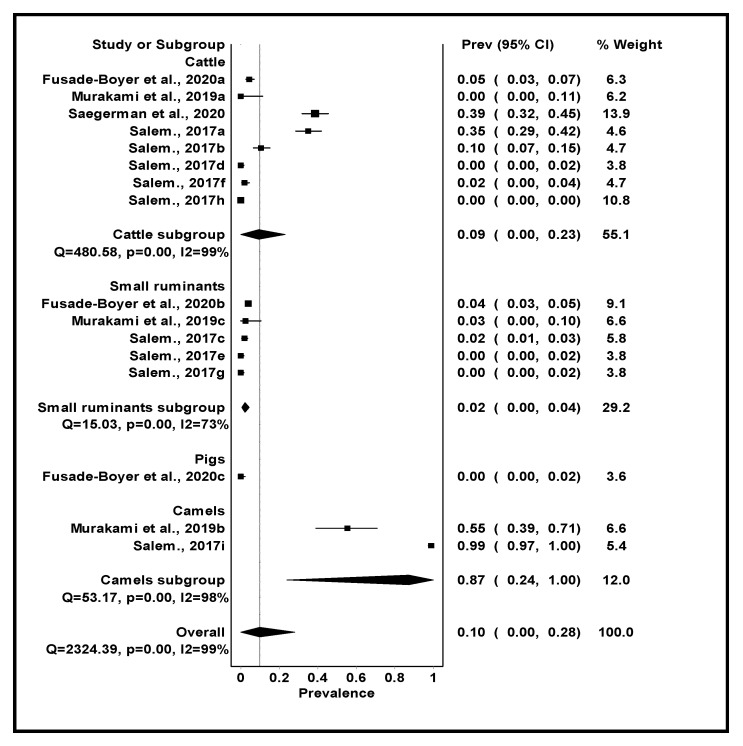
Forest Plot of the Seroprevalence of IDV according to subgroups.

**Table 1 viruses-13-02411-t001:** Distribution of influenza A virus and detection of antibodies in non-human mammalian hosts in Africa.

Region	Country	Influenza A Virus	Influenza A Virus Antibodies	Host Species	Reference
**African Island**	Reunion Island	H1N1pdm09	H1N1, H1, H3	Pigs	[62]
**Central Africa**	Cameroon	H1N1pdm09	H1N1pdm09, H1N2, H3N2	Pigs	[63,64,65]
**East Africa**	Ethiopia	ND ^1^	None ^2^	Horses	[66]
	Kenya	H1N1pdm09, IAV H1N1, H3N2 None ^2^ IAV ^3^ ND ^1^	H1N1, H3N2, IAV ND ^1^ IAV ^3^ H1N1 EIV ^4^	Pigs Olive baboons Cats Dogs Camels	[48,67,68] [69] [48] [48] [58]
	Uganda	IAV ^3^	IAV ^3^, H1	Pigs	[70,71]
**North Africa**	Egypt	H3N8, H7N7, H5N1 H1N1pdm09, H3N2, H5N1, H9N2 ND ^1^ ND ^1^ H9N2-like virus	H3N8, H7N7, H5N1 H1N1, H1N1pdm09, H5N1, H5N2, H5, H9 H5N1 None ^2^ ND ^1^	Equids Pigs Cats, Dogs, Rats Buffaloes, Cattle, Goats, Sheep Bats	[52,53,59] [51,72,73] [51] [51] [61]
	Morocco	ND ^1^	H3N8, H7N8	Equids ^5^	[54]
	Sudan	ND ^1^	EIV ^4^	Equids ^5^	[55]
	Tunisia	ND ^1^	EIV ^4^	Horses	[56]
**Southern Africa**	Namibia	ND ^1^ ND ^1^ ND ^1^ ND ^1^ ND ^1^	H1, H5 H4, H11 H1, H3, H5, H7, H8, H9, H11, H13, H14, H16 H7 H1	Black Rhino Wildebeest Caracals Honey Badger Lion	[28] [28] [28] [28] [28]
**West Africa**	Benin	None ^2^	ND ^1^	Pigs	[74]
	Burkina Faso	None ^2^	H1N1, H1N1pdm09	Pigs	[75]
	Côte d’Ivoire	None ^2^	None ^2^	Pigs	[74]
	Ghana	H1N1pdm09 ND ^1^	H1N1pdm09, H3N2 H3, H5, H8, H9, H12	Pigs Bats	[20,76] [60]
	Mali	ND ^1^	H3N8	Donkeys	[57]
	Niger	H3N8	ND ^1^	Donkeys, Horses	[77]
	Nigeria	H1N1, H3N2, H5N1, H1, H3, H5 H3N8 None ^2^	H1N1, H1N1pdm09, H3N2, H5N1, H3, H7, IAV ^3^ ND ^1^ IAV ^3^, H3N8	Pigs Donkeys, Horses Dogs	[65,76,78,79,80,81,82,83,84,85,86,87] [88] [49,50]
	Senegal	H3N8	ND ^1^	Donkeys, Horses	[77]
	Togo	H1N1pdm09	ND ^1^	Pigs	[89]

^1^ ND–Not Done; ^2^ None–Investigated but not detected; ^3^ IAV–Influenza A Virus (IAV–matrix gene detected but not subtyped; IAV antibodies–used multispecies ELISA kit); ^4^ EIV–Equine influenza virus (subtype not specified); ^5^ Equid-Horses, donkeys, mule.

**Table 2 viruses-13-02411-t002:** Distribution of influenza D virus and their antibodies in non-human mammalian species in Africa.

Region	Country	Influenza D Virus	Influenza D Virus Antibodies	Host Species	Reference
**East Africa**	Ethiopia	ND ^1^	IDV	Camels, Goats	[40]
	Kenya	ND ^1^	IDV	Camels	[40]
**North Africa**	Morocco	ND ^1^	IDV	Cattle	[40,90]
**West Africa**	Benin	ND ^1^ ND ^1^	IDV None ^2^	Cattle Sheep, Goat	[40] [40]
	Togo	None ^2^ ND ^1^ None ^2^	IDV IDV None ^2^	Cattle, Smail ruminants Cattle, Goats, Sheep Pigs	[91] [40] [91]

^1^ ND–Not done; ^2^ Not detected.

**Table 3 viruses-13-02411-t003:** Estimated pooled prevalence and seroprevalence of IAV in pigs in Africa.

IAV	Regions	Sample Size	No. Positive	Pooled Prevalence/Seroprevalence (%)	95% CI ^1^	*I*^2^ (%)
Overall Prevalence	Africa	10,703	370	1.6%	0–55	98
	African Islands	474	62	13.2	10–16	-
	Central Africa	104	2	2.4	0–6	-
	East Africa	5196	23	0.3	0–1	91
	North Africa	433	122	10.4	0–100	100
	West Africa	4496	161	2.2	0–5	98
Overall Seroprevalence	Africa	10,870	2095	14.9	5–28	99
	African Islands	1203	399	33.2	31–36	-
	Central Africa	98	27	27.8	19–37	-
	East Africa	5098	680	12.6	7–13	96
	North Africa	585	226	25.8	0–100	100
	West Africa	3886	763	14.9	0–41	99

^1^ CI–Confidence Interval.

**Table 4 viruses-13-02411-t004:** Estimated Pooled seroprevalence of IDV in non-human mammalian species in Africa.

Subgroup	Sample Size	No. Positive	Pooled Seroprevalence (%)	95% CI ^1^	*I*^2^ (%)
**Overall seroprevalence**	3992	536	9.9	0–28	99
**Cattle**	2260	190	9.3	0–23	99
**Small Ruminants**	1321	35	2.2	0–4	73
**Pigs**	80	0	0.0	0–2	-
**Camels**	331	311	87.2	24–100	98

^1^ CI–Confidence Interval.

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
