# Peer review of "Influenza A and D Viruses in Non-Human Mammalian Hosts in Africa: A Systematic Review and Meta-Analysis"

_viruses, 2021, doi:10.3390/v13122411_

Round 1

Reviewer 1 Report

    Influenza is an important disease of great concern, and current prevalence data in Africa are patchy and limited. This manuscript conducted a systematic review and meta-analysis to investigate the prevalence and current knowledge of influenza A virus (IAV) and influenza D virus (IDV) in non-avian animals in Africa. The result showed that the estimated pooled prevalence and seroprevalence of IAV in pigs in Africa was 1.6%  and 14.9%, respectively. The seroprevalence of IDV was 87.2%  in camels, 9.3% in cattle, 2.2%  in small ruminants and 0.0%  in pigs. H1N1 was still the most prevalent sorotype, and H5N1 was also detected in pigs. Other subtypes detected included H3N8 and H7N7 in equids, H1N1, H3N8 and H5N1 in dogs and cats. Furthermore, various wildlife animals were exposed to different IAV subtypes. The result may provide useful reference fro the prevention of influenza in Africa

Minor comments

  1. Table 3 and 4 can be merged; Figure 4 and 5 can be merged
  2. No Pooled prevalence data of IDV in non-avian species?
  3. If possible, please provide the genotype of HA gene of HPAIV H5N1, such as 2.3.4.4, in the discussion.

Author Response

Responses to Reviewers Comments

Reviewer 1

Minor comments

  1. Tables 3 and 4 can be merged; Figure 4 and 5 can be merged

Response: Tables 3 and 4 have been merged and figures 4 and 5 have been merged, (page 10-11)

  1. No Pooled prevalence data of IDV in non-avian species?

Response: There was no pooled prevalence data of IDV in non-avian species because most of the eligible studies for IDV performed serological prevalence apart from one study. Therefore, we could not pool the data for the only study.

  1. If possible, please provide the genotype of HA gene of HPAIV H5N1, such as 2.3.4.4, in the discussion.

Response: Genotype of HPAIV H5N1 has been added on page 13, lines 613-614 and page 14, line 657.

Reviewer 2 Report

In this review, Kolanda et al review literature and combine the findings from those papers about prevalence of influenza A and D viruses in non-avian hosts in Africa. While the topic is relevant and the discussion of the data found in the original research and summarized with the meta-analysis is needed, the paper is organized like an original research article, which puts forth a lot of wording that the readers do not need (review of literature in a review is implied). The manuscript needs to be rearranged and made more concise so that readers do not lose interest.

Specific comments

  1. The Methods section is too long and needs to be more concise. Much can be moved to supplemental material (like how the literature search was done). Other examples: Section 2.4 is not needed-the reader does not need to know Excel was used/titles were put in the file and this does not assist the reader with the summary of data.

  1. “Results” and “Discussion” titles should be removed (this is appropriately submitted as a review) and only descriptive titles used. Furthermore, the current “Discussion” should be combined with the different (current) 3.4-3.6 sections.

  1. Sections 3.1-3.3 Should all be combined into 1 heading and should be significantly shortened to be more concise. For example-stating the number of duplicate articles found. Additionally, Figure 1 provides a lot of what is needed for the reader.

Author Response

Responses to Reviewers Comments

Reviewer 2

In this review, Kolanda et al review literature and combine the findings from those papers about prevalence of influenza A and D viruses in non-avian hosts in Africa. While the topic is relevant and the discussion of the data found in the original research and summarized with the meta-analysis is needed, the paper is organized like an original research article, which puts forth a lot of wording that the readers do not need (review of literature in a review is implied). The manuscript needs to be rearranged and made more concise so that readers do not lose interest.

Specific comments

  1. The Methods section is too long and needs to be more concise. Much can be moved to supplemental material (like how the literature search was done). Other examples: Section 2.4 is not needed-the reader does not need to know Excel was used/titles were put in the file and this does not assist the reader with the summary of data.

Response: Thank very much for the comments. We have moved the description of the literature search to supplementary materials Protocol S1. We have also removed the sentence “A data extraction file was developed in MS Office Excel® 2016” under section 2.4. However, we have maintained the description of data extraction as we think it is important for readers to find this information within the manuscript for immediate appreciation of which data was extracted from the eligible papers without having to refer to the supplementary materials.

  1. “Results” and “Discussion” titles should be removed (this is appropriately submitted as a review) and only descriptive titles used. Furthermore, the current “Discussion” should be combined with the different (current) 3.4-3.6 sections.

Response: Thank you very much for the comment. Respectfully, we are of the view that these sections shoul be maintained. This is because our Systematic review and meta-analysis followed a PRISMA checklist (see attached Checklist S1 in supplementary materials) which requires that these sections be written separately (see also page 3 under materials and methods). Numerous systematic reviews and meta-analysis articles have these sections as evidenced from the following articles:

  1. Jakobsen, K.K.; Carlander, A.-L.F.; Bendtsen, S.K.; Garset-Zamani, M.; Lynggaard, C.D.; Grønhøj, C., et al. Diagnostic Accuracy of HPV Detection in Patients with Oropharyngeal Squamous Cell Carcinomas: A Systematic Review and Meta-Analysis. Viruses. 2021, 13, 1692, https://www.mdpi.com/1999-4915/13/9/1692.
  2. Marchello, C.S.; Hong, C.Y.; Crump, J.A. Global Typhoid Fever Incidence: A Systematic Review and Meta-analysis. Clin Infect Dis. 2019, 68, S105-s116, DOI: 10.1093/cid/ciy1094.
  3. Khan, S.U.; Anderson, B.D.; Heil, G.L.; Liang, S.; Gray, G.C. A Systematic Review and Meta-Analysis of the Seroprevalence of Influenza A(H9N2) Infection Among Humans. J Infect Dis. 2015, 212, 562-569, DOI: 10.1093/infdis/jiv109.
  4. Chatziprodromidou, I.P.; Arvanitidou, M.; Guitian, J.; Apostolou, T.; Vantarakis, G.; Vantarakis, A. Global avian influenza outbreaks 2010-2016: a systematic review of their distribution, avian species and virus subtype. Syst Rev. 2018, 7, 17, DOI: 10.1186/s13643-018-0691-z.

  1. Sections 3.1-3.3 Should all be combined into 1 heading and should be significantly shortened to be more concise. For example-stating the number of duplicate articles found. Additionally, Figure 1 provides a lot of what is needed for the reader.

Response: Thank you for the suggestions. We have merged sections 3.1 and 3.2 into section 3.1. Furthermore, we have deleted sentences that described article selection and just referred to Figure 1. However, we have maintained 3.3 as 3.2 now as we feel it describe a different subject from 3.1 and it may be better placed as a separate section.

Reviewer 3 Report

In their manuscript, “Influenza A and D Viruses in Non-avian Hosts in Africa: A Systematic Review and Meta-analysis”, Kalonda et al present their findings following a review of the literature relevant to influenza A and D occurrence in non-human mammalian species in Africa. They further conduct a meta-analysis on the occurrence of flu A in pigs and flu D in several species. The work is conducted well and the manuscript is well-written.

Major issues

  • It is advisable to conduct a search on the OIE WAHIS platform to check if outbreaks of the targeted viruses were reported among targeted species by African Countries. This is important as outbreaks detected by veterinary authorities do not typically get published in the peer-reviewed literature. However, this will be important so that a more accurate description of the situation is presented.

Minor issues

  • It will be better to change non-avian to non-avian and non-human species or non-human mammalian species to show that the work did not include studies of IAV or IDV in humans.
  • Though the time frame of study inclusion is sound, it will be better to add a phrase to the methods showing why the time period was limited to 2000-2020 and not earlier.
  • Line 414: please consider a better synonym to ‘intimates’

Author Response

Responses to Reviewer’s Comments

Reviewer 3

In their manuscript, “Influenza A and D Viruses in Non-avian Hosts in Africa: A Systematic Review and Meta-analysis”, Kalonda et al present their findings following a review of the literature relevant to influenza A and D occurrence in non-human mammalian species in Africa. They further conduct a meta-analysis on the occurrence of flu A in pigs and flu D in several species. The work is conducted well and the manuscript is well-written.

Major issues

  • It is advisable to conduct a search on the OIE WAHIS platform to check if outbreaks of the targeted viruses were reported among targeted species by African Countries. This is important as outbreaks detected by veterinary authorities do not typically get published in the peer-reviewed literature. However, this will be important so that a more accurate description of the situation is presented.
  •  

Response: We would like to thank the Reviewer for this suggestion. The OIE WAHIS was searched and four reports that met our inclusion criteria were included in this study. This has led to an increase in the number of articles/records and countries reporting the influenza virus in non-human mammalian hosts. Further, these reports have been incorporated and described on page 9 and in Table 1 on page 10.

Minor issues

  • It will be better to change non-avian to non-avian and non-human species or non-human mammalian species to show that the work did not include studies of IAV or IDV in humans.
  •  

Response: “Non-avian hosts” in the title has been changed to “non-human mammalian hosts” and this change has been effected throughout the manuscript.

  • Though the time frame of study inclusion is sound, it will be better to add a phrase to the methods showing why the time period was limited to 2000-2020 and not earlier.

Response: A sentence giving a reason for limiting our search to 2000-2020 has been added and reads “We searched literature from 2000-2020 because these are the two decades in which African countries reported a rise in influenza virus outbreaks in poultry and encompasses a period in which 2009 influenza pandemic (A(H1N1)pdm09) emerged.”. Kindly check supplementary materials S1 Protocol.

  • Line 414: please consider a better synonym to ‘intimates’

Response: “intimates” has been changed to “may suggest” on page 14, line 656.

Round 2

Reviewer 2 Report

This reviewer is satisfied with the changes the authors have made.